# Population-Based Analysis of Trends in Incidence and Survival of Human Papilloma Virus-Related Oropharyngeal Cancer in a Low-Burden Region of Southern Europe

**DOI:** 10.3390/ijerph19084802

**Published:** 2022-04-15

**Authors:** Jordi Rubió-Casadevall, Elna Ciurana, Montserrat Puigdemont, Arantza Sanvisens, Jordi Marruecos, Josefina Miró, Antoni Urban, Rosa-Lisset Palhua, Ferran Martín-Romero, Maria Rosa Ortiz-Duran, Rafael Marcos-Gragera

**Affiliations:** 1Medical Oncology Department, Catalan Institute of Oncology, Hospital Josep Trueta, 17007 Girona, Spain; 2Descriptive Epidemiology, Genetics and Cancer Prevention Group, Girona Biomedical Research Institute (IDIBGI), 17190 Girona, Spain; mpuigdemont@iconcologia.net (M.P.); asanvisens@iconcologia.net (A.S.); rmarcos@iconcologia.net (R.M.-G.); 3School of Medicine, University of Girona (UdG), 17004 Girona, Spain; elna.ciurana@gmail.com (E.C.); jmarruecos@iconcologia.net (J.M.); rortizduran.girona.ics@gencat.cat (M.R.O.-D.); 4Epidemiology Unit and Girona Cancer Registry, Oncology Coordination Plan Department of Health Government of Catalonia, Catalan Institute of Oncology, 17004 Girona, Spain; 5Radiotherapy Oncology Department, Catalan Institute of Oncology, Hospital Josep Trueta, 17007 Girona, Spain; 6Pathology Department, Girona Clinic, 17007 Girona, Spain; pmiroap@clinicagirona.cat; 7Pathology Department, Corporació de Salut del Maresme La Selva, Hospital Sant Jaume de Calella, 08370 Barcelona, Spain; aurban@salutms.cat; 8Pathology Department, Serveis de Salut Integrats Baix Empordà, Palamos Hospital, 17230 Girona, Spain; rpalhua@ssibe.cat; 9Pathology Department, Fundació Salut Emporda, Figueres Hospital, 17600 Girona, Spain; lfmartin@salutemporda.cat; 10Pathology Department, Catalan Institute of Health, Hospital Josep Trueta, 17007 Girona, Spain

**Keywords:** oropharyngeal cancer, epidemiology, survival, incidence, p16, human papilloma virus

## Abstract

**Introduction:** Human papilloma virus (HPV)-related oropharyngeal carcinoma (OPC) can be considered a new subtype of cancer with different clinical characteristics and prognosis than that related to tobacco. Its incidence is increasing worldwide. Its epidemiology has been widely studied in areas such as North America and Northern Europe, but less is known in Southern Europe. **Methods:** We analyzed the epidemiology of OPC using the database from Girona’s population-based Cancer Registry, in the North-East of Spain, from 1994 to 2018. To analyze differences between neoplasms related to human papillomavirus or not, we determined the immunohistochemical expression of p16 in cases within four time periods: 1997–1999, 2003–2005, 2009–2011, and 2016–2018. **Results:** Oropharyngeal cancer incidence increased significantly from 2001 to 2018 with an Annual Percentage of Change (APC) of 4.1. OPC p16-positive cases increased with an APC of 11.1. In the most recent period, 2016–2018, 38.5% of OPC cases were p16-positive. European age-standardized incidence rate was 4.18 cases/100.000 inhabitants-year for OPC cancer and 1.58 for those p16-positive. Five-year observed survival was 66.3% for p16-positive OPC and 37.7% for p16-negative. **Conclusions:** Although with lower burden than in other regions, p16-positive oropharyngeal cancer is increasing in our area and has a better prognosis than p16-negative OPC.

## 1. Introduction

Human Papillomavirus (HPV), a sexually transmitted infection which is recognized as one of the major causes of various infection-related cancers, has been rising in importance in some locations of head and neck cancer. Since the beginning of the 21st century, the etiological role of HPV in oral and more predominantly oropharyngeal cancer has been defined [1] and widespread oral sexual practices are a well-known risk factor [2]. Despite most of HPV infections resolving spontaneously, likewise not progressing to cancer, it has been proven that persistent infection with high-risk types of HPV cause carcinogenic lesions [3].

The changes in smoking habits and the arising role of HPV as a carcinogen has produced a shift in epidemiology of head and neck cancer (HNC): while overall HNC incidence has decreased, oropharyngeal cancer (OPC) incidence has sharply increased, giving rise to an epidemic of a new subtype of OPC related to HPV [4], with a different clinical and prognostic profile than those which are tobacco-related [5].

The emergence of this entity, that characterizes OPC in contrast to other head and neck locations [6], has been widely studied in regions with a high burden of oral HPV infection, such as North America and Northern Europe, but less is known about the onset of HPV-related OPC in Southern Europe, especially from population-based data [7]. Our aim was to analyze the trend of the incidence and survival of HPV-related OPC from an epidemiological point of view, using data from a Cancer Registry, thus avoiding biases that may have arisen from hospital centers.

## 2. Methods

We analyzed the database from the Girona Cancer Registry (GCR), a population-based cancer registry in Girona province, located in the North-East of Spain, which started registration in 1994. The region has seven community hospitals and a reference-center, the University Hospital Josep Trueta. The population covered was 747.157 inhabitants according to the census on 1 January 2018 (www.idescat.cat, accessed on 31 January 2022). At GCR, cases are registered according to the International Agency for Research on Cancer (IARC) guidelines with a completeness of 96.3%. The International Classification for Diseases—Oncology third edition (ICD-O-3) is used to register cases [8]

For the analysis of the incidence of all head and neck cancer, we selected primary tumors diagnosed from 1994 to 2018 in topographical sites C00–C14 and C30–C32. OPC was considered as those cases with topographical sites C01.9 (base of the tongue), C05.1–C05.2 (soft palate and uvula), C09.0–C09.9 (palatine tonsils), and C10.0–C10.9 (oropharynx).

To perform the analysis of p16 expression, we selected a cohort of oropharyngeal cancer cases diagnosed in the time periods 1997–1999, 2003–2005, 2009–2011, and 2016–2018.

All tumors selected were squamous cell carcinomas, corresponding to ICD-O-3 morphological codes 8070–8072.

Determination of immunohistochemical expression of p16INK4a antibody was performed on formalin-fixed, paraffin-embedded tissue section, in the respective pathology department. A moderate to strong nuclear and cytoplasmic staining in ≥70% of the tumor was categorized as p-16 positive. Nuclear and cytoplasmic staining in <50% of the tumor was categorized as p16-negative. A nuclear and cytoplasmic staining in ≥50% but <70% of the tumor was suitable for an HPV DNA-based test, but it was not needed in any of our cases.

We considered missing cases those in which a paraffin-embedded tissue was not obtained and did not evaluable cases were those in which the tissue to perform p16 analysis was not sufficient.

Incidence was analyzed in terms of crude rate (CR) and World and European Age Standardized Incidence Rate (ASIRw, ASIRe). Trends were assessed using the estimated Annual Percentage of Change (APC) of the ASRe. The Joinpoint log-linear regression analysis version 4.3.2.1.0 model was used carried out to calculate APC, with a 95% confidence interval (CI) using Joinpoint Regression Program version 4.9.0.0 (Statistical Methodology and Applications Branch, Surveillance Research Program, National Cancer Institute, Bethesda, MD, USA). The Chi-square test and t-test were used in qualitative and quantitative variables to analyze differences between p16-positive and p16-negative characteristics.

Follow-up time was calculated from diagnosis to patients’ last known vital status, obtained by means of record linkage to the Mortality Registry of Catalonia and the Spanish National Death Index, with 31 December 2021 as the most recent update. We estimated five-year net survival using the cohort approach for patients diagnosed in the three first periods and the period approach for patients diagnosed during 2016–2018, as five years of follow-up data were not available for all patients. Observed survival (OS) and net survival (NS) were estimated using Kaplan–Meier and Pohar–Perme methods, respectively. For data analysis, R software (version 3.6.1) and Stata software (version 11.1, StataCorp LLC, College Station, TX, USA) were used. Statistical significance was defined as two-sided *p* < 0.05.

## 3. Results

In the GCR database from the 1994–2018 period, 3257 cases of HNC were identified, 2780 males (85.3%) and 477 females (14.6%). Mean age at diagnosis was 64.1, standard deviation (SD) ± 12.7.

Regarding OPC, 509 cases were identified in the CRG database from the 1994–2018 period. A number 435 males (85.5%) and 74 females (14.5%) were included. Mean age at diagnosis was 60.9 (SD ± 11.2).

The distribution according to topographical site of the whole cohort of OPC was 118 cases (23.2%) in the base of tongue, 44 cases (8.6%) in uvula and soft palate, 211 cases (41.4%) in palatine tonsil, and 136 cancers (26.7%) registered as oropharynx.

Mean age at diagnosis was 57.8 years (SD 14.0) in the 1997–1999 period, 59.0 years (SD:11.7) in the 2003–2005 period, 61.8 years (SD:10.9) in the 2009–2011 period, and 63.3 years (SD:10.3) in the 2016–2018 period.

Incidence for all head and neck cancer and oropharyngeal cancer are summarized on Table 1.

To assess the trends of incidence for all head and neck cancer, the Joinpoint analysis of ASIRe_,_ computed to assess specific turning points, and resulted in a significant decrease in incidence from 1994 to 2010 (Figure 1), with an APC of −1.8 (95% CI: −2.5; −1.0) and a non-significant increase in incidence from 2010 to 2018 with an APC of 0.5 (95% CI: −1.6; 2.7).

During the period 1994–2018, OPC incidence had a non-significant increase in both sexes. An overall APC of 1.2 (95% CI: −0.5; 2.7) was obtained. Nonetheless, the Joinpoint analysis of ASIRe, computed to assess specific turning points in trends, resulted in a significant increase in incidence from 2001 to 2018 (Figure 2), with an APC of 4.1 (95% CI: 1.6; 6.7), but a non-significant decrease in incidence from 1994 to 2000 with an APC of −8.7 (95% CI: −16.8; 0.2).

In the GCR database from the selected periods of 1997–1999, 2003–2005, 2009–2011, and 2016–2018, 245 cases of OPC were identified. We obtained: 54 p16-positive cases (22.0%), 155 p16-negative cases (63.3%), 28 missing cases (11.4%), and 8 non-evaluable cases (3.3%). The distribution of missing plus not evaluable cases for each time period was 38.6% for 1997–1999, 18.7% for 2003–2005, 7.7% for 2009–2011, and 5.7% for 2016–2018. Finally, the analysis of p16 expression was performed on 209 of the 245 cases, meaning 85.3% of the total cohort. We show this data in Table 2.

In Table 3 we summarize the characteristics of p16-positive and -negative cases according to sex, median age at diagnosis, sublocation in oropharynx defined by ICD-O-3 site, and calendar period of diagnosis.

The observed percentage of HPV-related OPC cases among all OPC cases in our series of 209 cases in which p16 expression analysis was performed, was 11.1% in the period 1997–1999 (3 out of 27), 17.9% in 2003–2005 (7 out of 39), 20% in 2009–2011 (12 out of 60), and 38.5% in 2016–2018 (32 out of 83 cases).

Crude and adjusted incidence rates of p16-positive and p16-negative cohorts in the calendar periods 1997–1999, 2003–2005, 2009–2011, and 2016–2018 are shown in Table 4.

In the analysis of ASIRe trends, we have observed a statistically significant decrease in the incidence of overall HNC from 1994 to 2010, but a non-significant increase in the period 2010–2018. Moreover, we found a significant increase in the incidence of overall OPC in the period 2001–2018, and a non-significant decrease from 1994 to 2001.

Focusing on the calendar periods when p16 analysis was performed, the p16-positive OPC incidence increased with an APC of 11.1 (CI 95%: 7.7; 14.6), while we obtained a non-significative increase in p16-negative OPC, with an APC of 2.01 (CI 95%: −1.0; 5.1), as shown in Figure 3.

The mean follow-up time was 4.9 years, with SD of 5.4 years.

Table 5 shows results in OS for the cohort of patients in which analysis of p16 immunostaining was performed. The long-rank test proves a significant difference in 5-year OS between the two cohorts. In the p16-positive cohort, 66.3% of the patients were still alive 5 years from diagnosis (Figure 4). Meanwhile, the probability of survival in the p16-negative cohort was much lower, at only 37.7% of patients alive 5 years from diagnosis.

Given that survival can also be influenced by the calendar period in which the patients were treated, we show OS and net survival (NS) between the different periods studied in Table 6, regardless of HPV status.

## 4. Discussion

In our cohort, the absence of a decrease in incidence observed in overall HNC from 2010 to 2018, in contrast to previous years, seems to be attributable to the increase of OPC found in period 2001–2018.

These changes in epidemiological trends could be attributed to the shift in OPC etiology, with declining importance for tobacco and alcohol in favor of HPV’s more important role. A reduction in tobacco consumption has been noticed in Spain according to the Spanish National Health Survey, as the prevalence of daily smoking declined from 42% in men and 27% in women in 2001 to 32% in men and 23% in women in 2011 (https://www.mscbs.gob.es/estadEstudios/estadisticas/encuestaNacional/aniosAnteriores.htm, accessed on 31 January 2022) and 26% in men and 19% in women in 2017 (https://ec.europa.eu › docs › 2019_chp_es_english, accessed on 31 January 2022). However, the detected non-significant increase in the HPV-unrelated cohort supports that there is still a dominant effect from smoking and that the impact of anti-tobacco measures has not yet been seen on this site of the head and neck sphere.

The incidence rates for the HPV-related OPC experienced a 2.5-fold increase from the third calendar period (2009–2011) to the fourth (2016–2018). These findings explain how changes in recent times regarding sexual behaviors, similar to tobacco habits, likely underlies the shift noticed in OPC etiology. Number of vaginal-sex and oral-sex partners has been associated with OPC risk [2,9]. An increase in number of sexual partners and a decrease in the median age at first sexual experience has been reported in Spain [10,11]. Thus, changes in sexual practices have potentially led to the increase in incidence rates of HPV-related OPC observed in our study. In any case, more knowledge is needed on the biology of the virus, since studies to detect the presence of HPV in the oropharynx of sexually active adolescents have not correlated this behavior with the sparse presence of HPV [12].

Focusing on the percentage of p16-positive cases in each calendar period, we observed a notable increase from 18.5% in 2009–2011 to 36.75% in 2016–2018. Globally, our results are not so far from other published in Spain. In a study performed with cases diagnosed between 1990 and 2009 in a single institution in Asturias, in the North of Spain, the p16-positive cases were 12% from a total of 248 OPC patients [13]. In a retrospective cohort study of all new patients diagnosed with primary OPC in four hospitals in Catalonia, North-Eastern Spain, from 1991 to 2013, a clear increasing trend in the risk of HPV positivity in OPC was found in the last 5-year period of those analyzed, 2012–2016. In it, 21.5% of patients were considered suffering from an HPV-related cancer, defined as HPV-DNA testing and p16^INK4a^ expression-positive [14]. Using both identification methods is more accurate, since the number of p16-positive cases that would be HPV-DNA test-negative, and therefore not due to the virus, is established above 10% [15]. Another study performed in 102 OPC patients treated in a network of hospitals in Madrid, Spain, between 2000 and 2008, found 26.7% of those to be p16-positive [16].

Those percentages were lower than the 32.3% reported in Italy in the period 2000–2018 [17], the 55% reported in Denmark between 2000 and 2017 [18], the 55% reported in South Wales, United Kingdom (UK), in the period 2001–2006 [19], and the 71.7% in USA in the period 2000–2004 [4]. Furthermore, and in contrast to ours, most of these studies used double analysis of HPV etiology of OPC, with HPV-DNA test and p16 immunostaining.

Regarding specific population crude incidence rate, in Girona it was 1.43 for OPC HPV positive patients and 2.28 for HPV negative in 2016–2018. Those results were opposite to those observed in the USA from 2013 to 2014, whose incidence was 4.62 for HPV-positive OPC and 1.82 for HPV-negative OPC [20].

HPV-related OPC increased in the province of Girona approximately 20 years behind other countries such as the United States [2] or Norway [21]. This time lag is probably explained due to Spaniards not experiencing the same sexual revolution around the 60s as the previously mentioned regions, but rather having a later change in social acceptance of sexual behaviors in the 80s.

Spain has been considered a country with a medium/low incidence of oropharyngeal cancer [22,23] but with an increasing trend [24]. ASIR_W_ for OPC observed in our study was higher than those that had been estimated in 2018 for Southern Europe [14]. Even so, they were lower than the ones estimated for Northern/Eastern/Western Europe, as well as North America and Australia [10]. In our study, we confirm a significative trend in increasing incidence of OPC and HPV-related OPC, as has been described mainly in the USA and North-European countries [25,26,27,28].

Age at diagnosis was not found to be significantly different between HPV-related and HPV-unrelated OPC in our setting, across all calendar periods. The mean age of diagnosis of the two groups was similar, about 61 years of age. It is widely described that HPV-related OPC affects younger patients than HPV-unrelated OPC. This has also been described in a recent multi-hospital study in Spain [15] but not in another one, in which an older period was analyzed [13]. We hypothesized that the incidence of HPV-related OPC would be most pronounced in younger individuals in comparison to those with HPV-unrelated OPC, a characteristic attributable to changes in sexual norms and fewer tobacco-associated cancers in younger generations. However, this effect has still not been seen. Interestingly, the association found between HPV-related OPC and younger age observed in the Spanish study was only present with cases that had double positivity for p16 expression and HPV-DNA, and was otherwise absent in cases that showed positivity for p16 expression alone.

OPC incidence was substantially higher in men and HPV-unrelated OPC significantly more frequent in men, most likely due to the higher daily smoking prevalence observed in males when compared to females in Spain [29]. In contrast, HPV-related OPC was diagnosed in a high percentage of women when compared to HPV-unrelated OPC, but it was still a man’s disease. This greater impact of HPV-related OPC in males is likely explained by higher oral HR HPV infection rates in males versus females. Difference in sexual behaviors by sex has been reported among Spaniards, in which males practice oral sex in a higher proportion and a have a higher average number of lifetime sexual partners in comparison to females [30,31].

In Spain, in the multi-center study performed from 1991 to 2016, only 11.1% of OPC were diagnosed in women and 21.9% were HPV-related [15]. In another multi-hospital study performed in Spain, 15% of OPC occurred in females, and 22.2% of those were HPV-related [16]. Those results are closer to the 28.3% of HPV-related OPC in women we obtained, and even to those obtained in a large study in Denmark, where HPV-related cases in women accounted for 24% [18], or 24.9% in The Netherlands [26]. Yet, significantly different from ours are the results described in the Italian study, in which 26.9% of the OPC diagnosed between 2000 and 2018 were in women and 51,4% of those were HPV-related [17]. Furthermore, 52.9% of OPC affecting women were HPV-related in the UK [19]. These geographical differences in the percentage divergence between genders can be influenced by the distribution in smoking habits.

As it has been known for a long time [5], our series confirm a better observed survival in HPV-related OPC in comparison to HPV-unrelated OPC (5-year OS: 64.8% versus 43.4%), as has already been reported in other regions [4,17,19]. This better outcome might also be affected by several factors, such as tobacco use and alcohol consumption, social economic status, performance status, comorbidities, disease stage, and treatment received among others. In our opinion, the differences in survival observed according to the period of years analyzed, with a better prognosis in the most recent years, are due to the improvement in radiotherapy techniques or the widespread use of chemotherapy that have been incorporated into the therapeutic arsenal. In addition, the survival of p16-positive cases from the last period is notably higher, although they also have a shorter follow-up time.

We found that the survival of p16-negative OPC patients was worse in the period 2009–2011 than in 2003–2005, although the opposite would be more understandable due to improvement in treatment techniques. We believe it is due to a number of cases with a wide confidence interval. Women with p6-negative OPC have worse survival than men probably due to small number of cases.

As our study is not a multivariant analysis of prognostic variables, we did not assess the involvement of the mentioned possible confounding factors other than HPV status in survival.

Our study has some limitations. First, we used only p16 immunostaining to consider an HPV-related cancer, assuming that up to 10% of positive cancers may not be due to papillomavirus [15].

Second, being a retrospective study using very long periods of time, we have many missing cases, especially in older years. In any case, we are proud to have been able to analyze more than 90% of the cohort in the most current periods.

Third, misclassification of cancer type by topography could have been a bias in our study, as in all other studies about epidemiology of head and neck cancer. It is inherent to the methodology of codification of tumors in ICD-O-3. The anatomic proximity between the oropharynx and other surrounding topographical sites often results in erroneous classification of this type of cancer, which depends on the information received by the professional who registers the case. We reviewed all cases included in our study and misclassified ones were reclassified, but we cannot be completely sure of having lost some cases. In addition, the clinical classification is not clear in most patients, especially on those tumors in the retromolar trigone that can be misclassified as tonsillar pilar, those in vallecula that can be misclassified in epiglottis, or the base of tongue, misclassified to other parts of it. Errors can be both ways and are often not cleared up by clinicians.

Finally, in GCR, we collected the demographic data of patients and histological characteristics of tumors, and we were not able to know clinical information besides age and sex. We did not register smoking habits or alcohol use, which could have been helpful for confounder adjustment of our results. Nonetheless, establishing relationships through a study of variables was not the aim of this study.

The increase in the incidence of HPV-related OPC underlines the crucial importance on the development of preventive strategies. The study of epidemiological distribution of its burden and its trends is critical for the implementation of efficacious public health interventions, as some strategies may be inappropriate in countries with lower disease burden and delayed increasing trends such as Southern Europe. On the one hand, HPV infection is potentially preventable by HPV vaccination, and consequently so are many of the HPV-related neoplasms [21]. Thus, introduction of males into national HPV vaccination programs, as has been done in Denmark or in the United Kingdom, must be considered as this upholds another horizon of effective prevention. On the other hand, health promotion and education of safe sex is essential to reverse the HPV-related, increasing OPC epidemic. In addition, improvements in HPV-related OPC screening techniques, such as saliva viral HPV DNA detection or circulating cell-free DNA or improvement of endoscopic imaging techniques for pre-malignant lesions opens the door to analyze the effectiveness of developing primary prevention programs for this type of tumor [32].

## 5. Conclusions

To our knowledge, this is the first population-based assessment of trends in incidence and survival of HPV-related OPC in Southern Europe. Our results indicate that epidemiological trends have begun to change similarly to those described in other geographical areas, but with a decalage of two decades. Meanwhile there has been a decrease in the incidence of overall head and neck cancer, and a significant increase in the incidence of overall oropharyngeal cancer from 2001 to 2018 has been noticed, due to the rising incidence of HPV-related carcinoma, sharply marked in the most recent years. In addition, better survival has been observed in HPV-related OPC in comparison to HPV-unrelated OPC in our area, as has also been reported in other countries.

## Figures and Tables

**Figure 1 ijerph-19-04802-f001:**
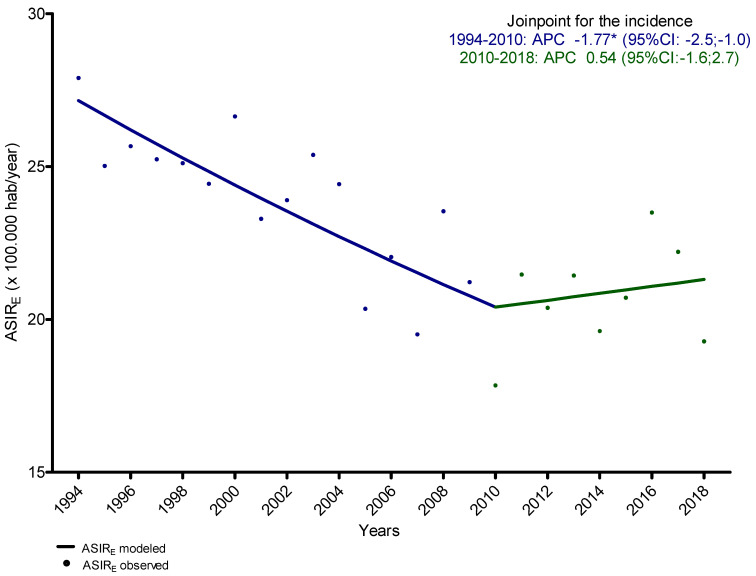
Trends in incidence of overall head and neck cancer, from 1994 to 2018. Joinpoint analysis of age-standardized to the European population incidence rates (ASIR_E_) assessing specific turning points. APC: Annual Percentage Change. * statistically significant.

**Figure 2 ijerph-19-04802-f002:**
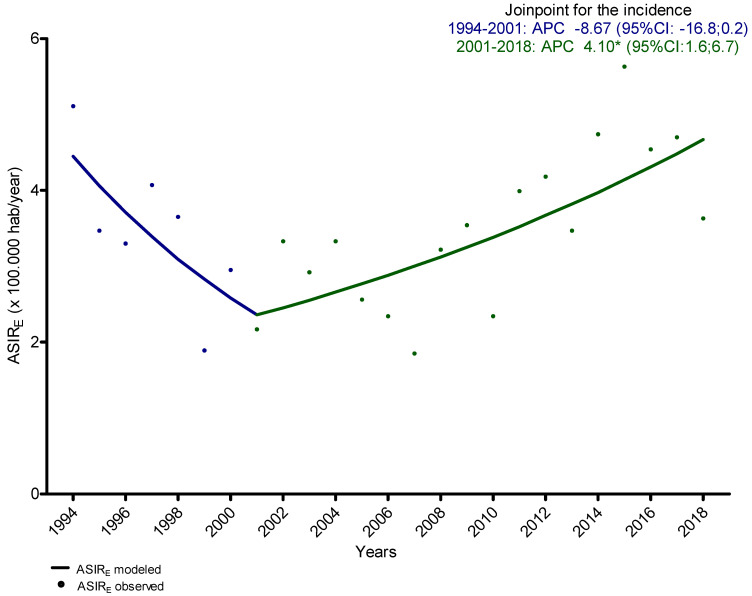
Trends in incidence of overall oropharyngeal cancer, from 1994 to 2018. Joinpoint analysis of age-standardized to the European population incidence rates (ASIR_E_) assessing specific turning points. APC: Annual Percentage Change. * statistically significant.

**Figure 3 ijerph-19-04802-f003:**
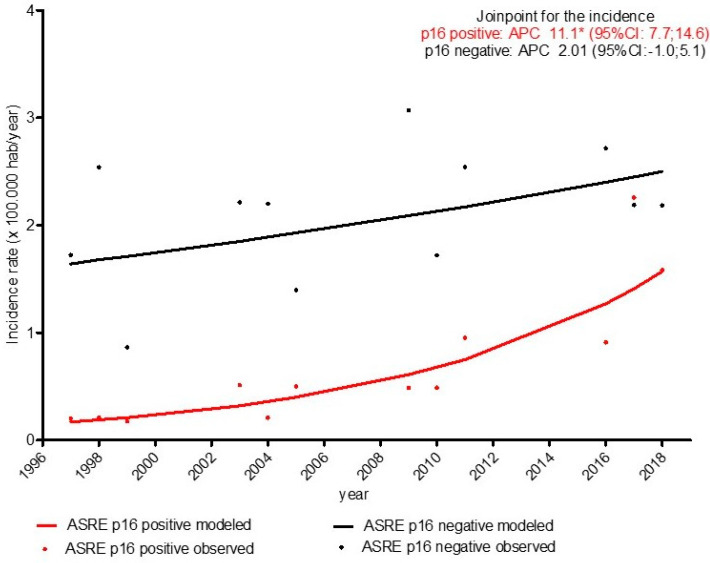
Trends in incidence of overall oropharyngeal cancer according to p16 expression. * statistically significant.

**Figure 4 ijerph-19-04802-f004:**
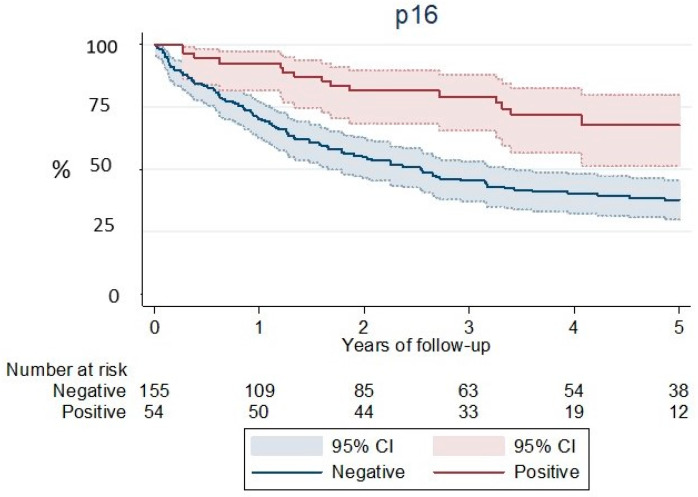
Observed survival according to p16 expression. Log rank test *p* < 0.001.

**Table 1 ijerph-19-04802-t001:** Incidence rates of overall head and neck cancer and oropharyngeal cancer between 1994 and 2018 according to sex.

	CR (95% CI)	ASIR_E_ (95% CI)	ASIR_W_ (95% CI)
ALL HEAD AND NECK			
Males	33.94 (32.68–35.20)	40.34 (38.84–41.91)	21.32 (20.50–22.18)
Females	5.84 (5.32–6.37)	6.03 (5.50–6.61)	2.94 (2.64–3.27)
Both sexes	19.91 (19.23–20.6)	22.37 (21.61–23.16)	11.96 (11.53–12.42)
OROPHARYNGEAL CANCER			
Males	5.31 (4.81–5.81)	6.13 (5.56–6.75)	3.50 (3.17–3.88)
Females	0.91 (0.7–1.11)	1.00 (0.79–1.26)	0.59 (0.46–0.78)
Both sexes	3.11 (2.84–3.38)	3.48 (3.19–3.80)	2.04 (1.86–2.24)

CR: crude rate; CI: confidence interval; ASIR_E_: age-standardized incidence rate to the European population. ASIR_W_: age-standardized incidence rate to the world population.

**Table 2 ijerph-19-04802-t002:** Cases evaluated or not for p16 expression, according to calendar period.

	Total Cases	Not Evaluated *	Evaluated
Period	n (%)	n (%)	n (%)
1997–1999	44 (17.9)	17 (38.6)	27 (61.4)
2003–2005	48 (19.6)	9 (18.7)	39 (81.3)
2009–2011	65 (26.5)	5 (7.7)	60 (92.3)
2016–2018	88 (35.9)	5 (5.7)	83 (94.3)
All periods	245 (100)	36 (14.7)	209 (85.3)

n: absolute number of cases. * Not evaluated: cases in which a paraffin-embedded tissue was not obtained, or it was not evaluable.

**Table 3 ijerph-19-04802-t003:** Characteristics of cases included in the p16 expression-based analysis.

	Total (245 Cases)	P16-Negative (155 Cases)	P16-Positive (54 Cases)	*p*-Value
	n (%)	n (%)	n (%)	
**Sex**				**0.003**
Males	209 (85.3)	138 (89.0)	39 (72.2)	
Females	36 (14.7)	17 (11.0)	15 (27.8)	
**Age**				0.578
Median [IQR]	59 [53–69]	59 [53–69]	62.5 [54–70]	
Mean ± SD	61.1 ± 11.6	61.1 ± 11.5	62.1 ± 11.2	
Minimum, Maximum	33, 93	39, 87	33, 84	
**Topographical site**				**0.012**
Base of Tongue (C01.9)	50 (20.4)	28 (18.0)	12 (22.2)	
Soft Palate (C05.1, C05.2)	28 (11.4)	22 (14.2)	1 (1.8)	
Palatine Tonsil (C09)	98 (40.0)	57 (36.8)	30 (55.6)	
Oropharynx (C10)	69 (28.2)	48 (31.0)	11 (20.4)	
**Period**				**0.006**
1997–1999	44 (18.0)	24 (15.5)	3 (5.6)	
2003–2005	48 (19.6)	32 (20.6)	7 (13.0)	
2009–2011	65 (26.5)	48 (31.0)	12 (22.2)	
2016–2018	88 (35.9)	51 (32.9)	32 (59.2)	

IQR: interquartile range; SD: standard deviation; N/n: absolute number of cases. p-value indicates differences between p16-positive and p16-negative patients; those in bold are statistically significant.

**Table 4 ijerph-19-04802-t004:** Incidence rates of p16-positive and p16-negative oropharyngeal cancer, stratified by calendar period.

	CR (95% CI)			ASIR_E_ (95% CI)			ASIR_w_ (95% CI)		
Period	p16+	p16−	Total (1)	p16+	p16−	Total (1)	p16+	p16−	Total (1)
1997–1999	0.19 (0.0–0.40)	1.49 (0.89–2.08)	2.73 (1.92–3.53)	0.20 (0.04–0.76)	1.70 (1.08–2.63)	3.19 (2.30–4.38)	0.13 (0.02–0.71)	1.05 (0.65–1.79)	1.98 (1.40–2.86)
2003–2005	0.37 (0.10–0.64)	1.69 (1.11–2.28)	2.54 (1.82–3.26)	0.41 (0.16–0.89)	1.92 (1.31–2.75)	2.87 (2.11–3.84)	0.24 (0.09–0.72)	1.17 (0.79-1.82)	1.77 (1.29-2.50)
2009–2011	0.54 (0.23–0.84)	2.14 (1.54–2.75)	2.90 (2.20–3.61)	0.64 (0.33–1.15)	2.44 (1.79–3.26)	3.35 (2.58–4.30)	0.37 (0.19–0.77)	1.39 (1.01-1.96)	1.88 (1.43-2.51)
2016–2018	1.43 (0.93–1.92)	2.28 (1.65–2.90)	3.93 (3.11–4.75)	1.58 (1.07–2.25)	2.37 (1.76–3.14)	4.18 (3.35–5.18)	0.85 (0.57–1.35)	1.29 (0.94-1.84)	2.27 (1.81-2.94)

(1) Total incidence rates have been analyzed used p16-positive, -negative, and missing cases. CI: confidence interval. CR: crude rate. ASIR_E_: age-standardized incidence rate to the European population. ASIR_W_: age-standardized incidence rate to the World population.

**Table 5 ijerph-19-04802-t005:** Five-year observed survival rates according to p16 expression, stratified by sex.

	% Males (95% CI)	% Females (95% CI)	% Both Sexes (95% CI)
p16-positive	64.6 (42.8–79.8)	70.0 (37.1–87.9)	66.3 (48.9–79.0)
p16-negative	40.7 (32.2–48.9)	12.6 (1.7–34.8)	37.7 (29.9–45.5)
All	47.9 (40.7–54.7)	41.5 (23.7–58.3)	47.0% (40.4–53.4)

CI: confidence interval. 5-years observed survival in all cases has been analyzed used p16-positive and -negative and missing cases.

**Table 6 ijerph-19-04802-t006:** Five-year observed and net survival rates according to p16 expression, stratified by period.

	P16-Positive		P16-Negative		All	
	OS (95% CI)	NS (95% CI)	OS (95% CI)	NS (95% CI)	OS (95% CI)	NS (95% CI)
1997–1999	33.3 (0.9–77.4)	34.1 (10.8–77.5)	12.5 (3.1–28.6)	15.3 (4.1–33.2)	29.5 (17.0–43.2)	39.3 (22.8–55.5)
2003–2005	57.1 (17.2–83.7)	66.6 (15.8–91.5)	56.2 (37.6–71.3)	60.4 (39.6–75.9)	56.2 (41.2–68.9)	60.9 (44.1–74.0)
2009–2011	58.3 (27.0–80.1)	62.3 (27.4–84.1)	35.4 (22.3–48.7)	37.2 (23.2–51.4)	43.1 (30.9–54.6)	46.2 (32.9–58.4)
2016–2018	82.8 (51.6–94.8)	88.1 (46.9–97.9)	50.1 (32.9–65.0)	53.2 (34.8–68.6)	63.1 (49.1–74.2)	67.3 (52.1–78.6)

CI: confidence interval. OS: observed survival; NS: net survival. OS and NS in all cases has been analyzed used p16-positive and -negative and missing cases.

## Data Availability

Data and materials are saved in the GCR and available for revision.

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
