# Peer review of "Population-Based Analysis of Trends in Incidence and Survival of Human Papilloma Virus-Related Oropharyngeal Cancer in a Low-Burden Region of Southern Europe"

_ijerph, 2022, doi:10.3390/ijerph19084802_

Round 1

Reviewer 1 Report

Thank you very much for the opportunity to review this study. The topic is very timely.

OPC incidence has increased over the last 20 years in several countries. HPV is an established cause of OPC (including the tonsil, base of the tongue, and other parts of the pharynx)whereas its etiologic role in OCC is unclear.

Manuscript requires minor corrections. Here are some minor suggestions for improvement.

INTRODUCTION

The introduction could be expanded with other considerations.

It is necessary to better investigate the epidemiology of Oropharyngeal Cancer (OPC) and the correlations with HPV. It is mentioned a new subtype of OPC related to HPV, with a different clinical and prognostic profile. It is necessary better clarify this point and add the references. When it was discovered?

I think it’s appropriate mentioned the role of the screening in this context as an important prevention role for this diasease. A good reference for this point are: “Baccolini V, Isonne C, Salerno C, Giffi M, Migliara G, Mazzalai E, Turatto F, Sinopoli A, Rosso A, De Vito C, Marzuillo C, Villari P. The association between adherence to cancer screening programs and health literacy: A systematic review and meta-analysis. Prev Med. 2022 Feb;155:106927. doi: 10.1016/j.ypmed.2021.106927. Epub 2021 Dec 23. PMID: 34954244; Timbang, M. R., Sim, M. W., Bewley, A. F., Farwell, D. G., Mantravadi, A., & Moore, M. G. (2019). HPV-related oropharyngeal cancer: a review on burden of the disease and opportunities for prevention and early detection. Human vaccines & immunotherapeutics.”

Methods and discussione are well structured.

Author Response

We sincerely appreciate the reviewer's comments.

As we are reminded, the knowledge of HPV as an etiological factor for oropharyngeal cancer has been known for decades. That is why we have expanded this information in the introduction ans supported it with a recent global review reference (page 2 lines 31-36)

We have also expanded the discussion focusing in the importance of screening and used one of the references suggested (page 16 lines 270-274)

Reviewer 2 Report

Overall, the authors conducted a retrospective study on a previous not studied population with statistical results to show the overall OPC as well as categorized according to P16 positive or negative in samples over twenty years.

The results section could use more details and the discussion section need to be rewritten/ organized, for example, using subtitles for each point would be helpful.  

Several points need to strength or clarify:  

  • A moderate to strong nuclear and cytoplasmic staining in ≥ 70% of 52 the tumor was categorized as p-16 positive. Nuclear and cytoplasmic staining in <50% of the tumor was categorized as p16-negative. “ A nuclear and cytoplasmic staining in ≥50% but <70% of the tumor was suitable for an HPV DNA-based test, but it was not needed in any of our cases.” The authors did not explain why? Is it because that they don’t really care whether these samples are HPV positive or not? This additional assay might change some statistical analysis outcomes.
  • Could author clarify Lines 83-86 “During the period 1994-2018, OPC incidence had a non-significant increase in both sexes. An overall APC of 1.2 (95% CI: -0.5; 83 2.7) was obtained. Nonetheless, the joint-point analysis of ASIRe computed to assess specific turning points in trends resulted in a significant increase in incidence from 2001 to 2018 (figure 2), with an APC of 4.1 (95% CI: 1.6; 6.7), but a non-significant decrease in incidence from 1994 to 2001 with an APC of -8.7 (95% CI: -16.8; 0.2).”
  • Fig 3 Red dot “ASIREP16 negative observed” should be “ASIREP16 positive observed”
  • Fix the label of Y axis of “Fig 4”
  • Line 161 “The incidence rates for the HPV-related OPC experienced a 2.5-fold increase from the third calendar period (2009-2011) to the 161 fourth (2016-2018).” Why do the authors think“ These findings explain how changes in recent times regarding sexual behaviors, similar to tobacco habits, likely underlies the shift noticed in OPC etiology”? Do you have any data to back up these claims in this population?
  • Line 185-187  The authors again explain the difference in the results between theirs and those reported by US group as “sex behavior/ revolution”. I hope there are some evidence to support this hypothesis.
  • Line 201 “Accordingly, these age discrepancies could be explained by a higher accumulation of mutations with age in the case of p16 positive/HPV-DNA negative patients, which we did not detect in our study as we used p16 positivity alone. Although p16-positive OPC is characterized by presenting at younger ages, the mean age at diagnosis in the different periods analyzed did not change substantially.” These two sentences are confusing.
  • Line 222-226 need references

Author Response

Answers to reviewer 2

We appreciate the comments from reviewer 2 that have helped us a lot to improve the manuscript. We answer them point by point

  • A moderate to strong nuclear and cytoplasmic staining in ≥ 70% of 52 the tumor was categorized as p-16 positive. Nuclear and cytoplasmic staining in <50% of the tumor was categorized as p16-negative. “A nuclear and cytoplasmic staining in ≥50% but <70% of the tumor was suitable for an HPV DNA-based test, but it was not needed in any of our cases.” The authors did not explain why? Is it because that they don’t really care whether these samples are HPV positive or not? This additional assay might change some statistical analysis outcomes.

In our series there were no cases with nuclear and cytoplasmic staining in ≥50% but <70% of the tumor. However, as we explained in line 251, in the discussion, not performing a DNA test can lead to a bias of overestimation of cases.

  • Could author clarify Lines 83-86 “During the period 1994-2018, OPC incidence had a non-significant increase in both sexes. An overall APC of 1.2 (95% CI: -0.5; 83 2.7) was obtained. Nonetheless, the joint-point analysis of ASIRe computed to assess specific turning points in trends resulted in a significant increase in incidence from 2001 to 2018 (figure 2), with an APC of 4.1 (95% CI: 1.6; 6.7), but a non-significant decrease in incidence from 1994 to 2001 with an APC of -8.7 (95% CI: -16.8; 0.2).”

In the incidence trend analysis for the entire period 1994-2018, an APC of 1.2 is obtained. When performing the joinpoint analysis, which detects points of change, two different periods were differentiated, 1994-2000 with one APC and 2001-2018 with another, as explained

  • Fig 3 Red dot “ASIREP16 negative observed” should be “ASIREP16 positive observed”

We have corrected this mistake

  • Fix the label of Y axis of “Fig 4”

We have fixed the label (%)

  • Line 161 “The incidence rates for the HPV-related OPC experienced a 2.5-fold increase from the third calendar period (2009-2011) to the fourth (2016-2018).” Why do the authors think “These findings explain how changes in recent times regarding sexual behaviors, similar to tobacco habits, likely underlies the shift noticed in OPC etiology”? Do you have any data to back up these claims in this population?

Although the literature on the evolution of sexual behavior in Spain is less than that published in the USA, we have provided references on how this evolution is taking place in our country. This, together with the consumption of tobacco that we also refer to, explains the trend of the incidence of this neoplasm. We also believe that our data and conclusions could be similar to other countries in Southern Europe with similar social characteristics.

  • Line 185-187 The authors again explain the difference in the results between theirs and those reported by US group as “sex behavior/ revolution”. I hope there are some evidence to support this hypothesis.

It is a subject of which there is no bibliography and we have not wanted to explain in the manuscript in more detail. But when the sexual revolution took place in the USA, in the 60's and 70's, in Spain we had a dictatorship and there was also a strong religious influence, so we experienced these changes in sexual freedom later.

  • Line 201 “Accordingly, these age discrepancies could be explained by a higher accumulation of mutations with age in the case of p16 positive/HPV-DNA negative patients, which we did not detect in our study as we used p16 positivity alone. Although p16-positive OPC is characterized by presenting at younger ages, the mean age at diagnosis in the different periods analyzed did not change substantially.” These two sentences are confusing.

We agree that sentence is confusing and is written to explain another study. Thus, we have removed it to simplify the message.

  • Line 222-226 need references

We do not have a reference for this topic. That is our opinion. We have changed the sentence to explain it as hyopthesis (lines 239-242 page 15)

Reviewer 3 Report

The manuscript reports on the Population-based analysis of trends in incidence and survival of human papilloma virus related oropharyngeal cancer in a low‐burden region of Southern Europe with the objective was to analyze the trend of the incidence and survival of HPV-related OPC from an epidemiological point of view.  Overall, the study is clearly explained and the results are presented concisely.

Specific comments

Page 2 line 30 – The introduction is too brief.  Eg. what is the significant of your study and it differs from others?

Page 1 line 23 – What is APC.  Please write in full when first time mentioned.

Page 3 line 64 – ...31 December 2021 as the…

Page 4 line 99 – Table 1…..1994 and 2018 according to sex.

Page 7 line 110 – What is the P-value as the author mentioned statistically significant.  P-value must also be mentioned in the text even it is described in the table or figures.

Page 10 line 138 -  What is 5-y?  Please write in full.

Author Response

Answers to reviewer 3

We sincerely appreciate the reviewer's comments, and we answer them point by point

Specific comments sent:

Page 2 line 30 – The introduction is too brief.  Eg. what is the significant of your study and it differs from others?

We have expanded the introduction by answering this question as well (page 2 line 45)

Page 1 line 23 – What is APC.  Please write in full when first time mentioned.

APC is “annual percentage of change”, we have written it in full in first time that is mentioned

Page 3 line 64 – ...31 December 2021 as the…

We have corrected this mistake

Page 4 line 99 – Table 1…..1994 and 2018 according to sex.

We have corrected this mistake

Page 7 line 110 – What is the P-value as the author mentioned statistically significant.  P-value must also be mentioned in the text even it is described in the table or figures.

We have explained this meaning better in table3. We have removed it from Table 2 for lack of meaning.

Page 10 line 138 -  What is 5-y?  Please write in full.

It means 5-years. We have written it in full

Reviewer 4 Report

Thank you for letting me review the proposed paper «Population-based analysis of trends in incidence and survival of human papilloma virus related oropharyngeal cancer in a low-burden region of Southern Europe». The introduction part is clear and well written, and the method part as well. The location of the tumours is very nicely described.

The results are interesting with trends in overall head and neck cancers decreasing, but the oropharyngeal rising (figure 1 and 2). The rents in incidence of overall oropharyngeal cancer according to p16 is not surprising, but still a very nice graph in figure 3. The time-intervals shows how p16-incidence has changed over time.

Table 5 shows a very low survival rate of p16 negative women, much lower than for the men. Could there be a reason for this? Could not seem to find this discussed in the discussion section.

In table 6 the p16 negative patients from the period 2003-2005 did much better then p16 negative patients from the other time-periods. The number of cases seem not to be very different from the period 1997-1999, but the 5-year survival is very much different. Any reason for this?

The discussion is good. Very nice to see the number of smokers decreasing so much over the last years.

Author Response

Dear reviewer

We sincerely appreciate your comments

Focusing on the aspects that we need to clarify, we believe that the worst survival in women may be because of the low number of cases, with a wide confidence interval. We added this in text (page 15, lines 245-246)

Regarding table 6, we confirmed after reviewing the data, that the survival of negative p16 is worse in the period 2009-2011 than in 2003-2005, although the opposite would be more understandable due to the improvement in treatment techniques. We believe it is a matter of number of cases, with a wide confidence interval. We explain it in the text (page 15, lines 243-245)